# The Characterization of Three Novel Insect-Specific Viruses Discovered in the Bean Bug, *Riptortus pedestris*

**DOI:** 10.3390/v14112500

**Published:** 2022-11-11

**Authors:** Chunyun Guo, Zhuangxin Ye, Biao Hu, Shiqi Shan, Jianping Chen, Zongtao Sun, Junmin Li, Zhongyan Wei

**Affiliations:** State Key Laboratory for Managing Biotic and Chemical Threats to the Quality and Safety of Agro-Products, Key Laboratory of Biotechnology in Plant Protection of MARA and Zhejiang Province, Institute of Plant Virology, Ningbo University, Ningbo 315211, China

**Keywords:** *Riptortus pedestris*, insect-specific virus, metatranscriptomic sequencing, small interference RNA

## Abstract

Insect-specific virus (ISV) is one of the most promising agents for the biological control of insects, which is abundantly distributed in hematophagous insects. However, few ISVs have been reported in *Riptortus pedestris* (Fabricius), one of the major pests threatening soybeans and causing great losses in yield and quality. In this work, field *Riptortus pedestris* was collected from six soybean-producing regions in China, and their virome was analyzed with the metatranscriptomic approach. Altogether, seven new insect RNA viruses were identified, three of which had complete RNA-dependent RNA polymerase (RdRp) and nearly full-length genome sequences, which were named Riptortus pedestris alphadrosrha-like virus 1 (RpALv1), Riptortus pedestris alphadrosrha-like virus 2 (RpALv2) and Riptortus pedestris almendra-like virus (RiALv). The three identified novel ISVs belonged to the family Rhabdoviridae, and phylogenetic tree analysis indicated that they were clustered into new distinct clades. Interestingly, the analysis of virus-derived small-interfering RNAs (vsiRNAs) indicated that only RiALv-derived siRNAs exhibited 22 nt length preference, whereas no clear 21 or 22 nt peaks were observed for RpALv1 and RpALv2, suggesting the complexity of siRNA-based antiviral immunity in *R. pedestris*. In conclusion, this study contributes to a better understanding of the microenvironment in *R. pedestris* and provides viral information for the development of potential soybean insect-specific biocontrol agents.

## 1. Introduction

The bean bug, *Riptortus pedestris* (Fabricius) (Hemiptera: Heteroptera), widely distributed in Asia, is a major agricultural pest of leguminous plants [1,2,3]. As a polyphagous insect, *R. pedestris* obtains nutrients and water by inserting its sucking mouthpart directly into plant tissues, including leaves, stems, flowers, pods and seeds, causing the partial necrosis of plant tissue and pathogen infection, which subsequently results in significant losses of yield and quality [4,5]. Although its occurrence has been found in flowering plants, barley, rice and fruit trees [6], further evidence suggests that the seeds or pods of leguminous plants are essential for their adult development and reproduction [4,7]. In recent years, *R. pedestris* has received increasing attention as the primary cause of soybean staygreen syndrome (Zhengqing, the Chinese common name), which is manifested as green leaves and stems, abnormal pods and seed abortion when most plants in the field have become fully mature [1,8]. Currently, Zhengqing has become the most epidemic and prominent issue in the Huang–Huai–Hai soybean-producing area of China [1], and the direct cause of Zhengqing is still highly debated. Although Zhengqing can be induced by pod removal or seed injury [8], recent studies demonstrated that a new geminivirus is also associated with soybean staygreen syndrome [9,10,11]. Additionally, field cage experiments demonstrated that *R. pedestris* infestation of soybean can clearly induce increased numbers of abnormal pods, aborted seeds and staygreen leaves and decreased yields [1,12]. Although several reports have indicated that Zhengqing is closely related to *R. pedestris* feeding, it remains unclear whether *R. pedestris* causes staygreen syndrome by direct feeding or by playing a role as a virus-transmitting vector, especially plant viruses. Thus, it is necessary to unveil the virome of *R. pedestris* so as to provide new insights into the mechanisms of insect-plant interactions.

Over the past decade, advances in high-throughput sequencing technologies have led to the discovery of a large number of RNA viruses, promoting their development in the field of invertebrate virology, mainly insects [13]. Viruses identified in insects have been recognized as insect-specific viruses (ISVs) that are confined exclusively to insects and are unable to replicate in vertebrate cells [14]. Most of the current identified ISVs belong to several specific viral taxa, including Rhabviridae, Flaviviridae, Baculoviridae and Parvoviridae [15,16]. The majority of ISVs are believed to establish close relationships with their host insects and considered as symbionts of host insects [17]. Numerous ISVs have been discovered in Hemiptera insects, especially the important plant virus vectors such as iflaviruses in planthopper [18], Flavivirus [19] and other families and even more unclassified viruses in aphids [20,21,22,23,24]. However, few ISVs have been reported in *R. pedestris*; and presently, the only ISVs reported in *R. pedestris* are Riptortus pedestris virus 1 (RiPV-1) and Riptortus pedestris virus 2 (RiPV-2), which were discovered from *R. pedestris* infected with the entomopathogenic fungus Beaveria bassiana [25,26]. RiPV-1 and RiPV-2 are positive-sense single-stranded RNA viruses belonging to the order Picornavirales. The genomes of both viruses are 9–11 kb in length and contain a large open reading frame (ORF) that encodes multiple conserved motifs, including helicase, RNA-dependent RNA polymerase (RdRP), etc. [25,26].

In this study, bean bug samples were collected from different regions in China, and the RNA virome was investigated by the metatranscriptomic technology. Altogether, three previously reported and seven novel insect RNA viruses were identified in *R. pedestris*. The three novel identified RNA viruses with almost complete genomes, Riptortus pedestris alphadrosrha-like virus 1(RpALv1), Riptortus pedestris alphadrosrha-like virus 2 (RpALv2) and Riptortus pedestris almendra-like virus (RiALv), were further characterized. Phylogenetic analysis suggested that the newly identified three viruses are the new members of the viral family Rhabdoviridae.

## 2. Materials and Methods

### 2.1. Field Insect Samples

Bean bug samples used in this experiment were collected from six soybean fields in two soybean-growing provinces (Fuyang, Suzhou, Hefei and Xuancheng of Anhui Province; Heze and Ji’ning of Shandong province,). The insects were sent to our laboratory alive, and some insects were transferred to trizol reagent (Invitrogen, Waltham, MA, USA) for RNA extraction; the remaining insect samples were reared separately in nylon mesh cages with cotton pads soaked in water and potted soybean plants (cv. Qihuang 34) in Ningbo University, Zhejiang.

### 2.2. RNA Extraction

Total RNA was extracted from insects using Trizol reagent (Invitrogen, USA) following the manufacturer’s instructions. Briefly, a pair of adult *R. pedestris* was randomly collected from the soybeans in each area, and then the whole bodies of the alive insects were moved into RNase-free microfuge tubes (2.0 mL). Subsequently, the extracted RNA samples were equally divided into two tubes for transcriptome and small RNA (sRNA) sequencing, respectively.

### 2.3. RNA-Sequencing

RNA samples were sent to Lc-Biotech (Hangzhou, China) for transcriptome sequencing. Briefly, an Illumina Novaseq™ 6000 (Illumina, San Diego, CA, USA) was utilized for RNA library (paired end, 150 bp ± 50 bp) construction and sequencing. After removing low-quality reads and adapter sequences, the clean data were de novo assembled using Trinity (version 2.8.5) with default parameters [27].

### 2.4. RNA Virus Discovery

Virus discovery was performed as previously described [28]. Briefly, each of the assembled RNA sequences was aligned against the NCBI viral RefSeq database using diamond BLASTx (E-value of ≤ 1 × 10 − 20) [29]. The viral homology contigs (≥2000 bp in length) were selected for further analysis and aligned to the entire NCBI nucleotide (NT) and nonredundant (NR) protein database to prevent false positive matches. Finally, RT-PCR was used to verify the identified virus-derived contigs. The primer sequences used are listed in Appendix A. PCR products were electrophoresed using 1% agarose gel (Yeasen Biotechnolog, Shanghai, China).

### 2.5. Genome Annotation and Transcript Coverage

The open reading frame (ORF) contained in each sequence was predicted by the ORF Finder online server (https://www.ncbi.nlm.nih.gov/orffinder, accessed on 30 June 2022). Meanwhile, the conserved domains were predicted by InterProScan (https://www.ebi.ac.uk/interpro/, accessed on 1 July 2022). To obtain viral transcriptome coverage data, the clean reads were mapped back to the viral genomes using Bowtie2 and Samtools [30,31].

### 2.6. Small RNA (sRNA) Sequencing and Analysis

sRNA sequencing was performed by Lc-Biotech (Hangzhou, China). A Small RNA v1.5 Sample Prep kit (Illumina, San Diego, CA, USA) was utilized to construct the sRNA libraries, which were later sequenced on the Illumina HiSeq 2500 platform. For sRNA analysis, the FASTX-Toolkit (version 0.0.14) was employed to remove the low-quality, large deletions and adapter sequences of raw reads, and the 18–30 nt long sRNA reads were extracted. Thereafter, those processed reads were matched to the assembled viral genome sequence (no zero matches were allowed) with Bowtie software [32]. Finally, custom perl scripts and Linux shell bash scripts were applied for providing a count estimate of virus-derived small interference RNAs (vsiRNAs).

### 2.7. Phylogenetic Analysis

The amino acid (aa) sequences of RdRp of the identified three novel ISVs, together with previously reported ISVs in the Rhabdoviridae retrieved from NCBI, were aligned by MAFFT (version 7.0) [33]. Moreover, Gblock was used to remove misaligned sequences [34] and select phylogenetically informative sites [35]. Thereafter, the optimal aa substitution model was evaluated by ModelTest-NG with default parameters [36]. Phylogenetic trees were then constructed by RAxML with maximum likelihood (ML) methods [37]. To check the reliability of those constructed trees, a 1000-replicate bootstrap was used in this analysis [35].

## 3. Results

### 3.1. Transcriptome Assembly and Virome Analysis in R. pedestris

A total of 1,268,096 contigs were generated from the de novo assembly of the clean RNA-seq reads (405,583,244), and details of the assembly can be found in Appendix A. The assembled sequences were later blast against the NCBI viral Ref-seq database to identify the potential viruses. As a result, 966 contigs showed high similarities to viral proteins (E-value ≤ 1 × 10 − 20). These contigs (length ≥ 2000 bp) were further searched against the NCBI NT and NR databases to confirm the viral-like sequences. It was observed from Table 1 that 13 viral contigs were identified in *R. pedestris*, representing 10 potential RNA viruses related to an order: Picornavirales and the following families: Iflaviridae, Rhabdoviridae, Permutotetraviridae and Flaviviridae (Table 1). Among them, 4 ISVs in *R. pedestris* collected from Anhui and Shandong provinces shared extremely high similarities to Riptortus pedestris virus-1 (RiPV-1) (≥99.0% aa identity, N = 3) and Riptortus pedestris virus-2 (RiPV-2) (≥98.9% aa identity, N = 1), annotated as unassigned insect RNA viruses found from bean bugs [25,26]. Another two ISVs in *R. pedestris* collected from Shandong Province had 94.64% and 94.75% identities with Guiyang srgiope bruennichi iflavirus 1, which belongs to the genus Iflavirus in the family Iflaviridae (Table 1). In addition to the known ISVs described above, seven novel viral-like contigs were also identified that exhibited low homology (≤45%) with known viruses (Table 1). The novel viruses identified in this study were named according to the isolated host and related family or genus, followed by a number, e.g., Riptortus pedestris Alphadrosrha-like virus 1. To further confirm the presence of these viruses in *R. pedestris*, the coverages of the 13 viral contigs were estimated by mapping clean-read sequences to the corresponding genome sequence. These transcripts were distributed throughout the entire viral sequence with high abundance at some loci, which might be the translation hotspots of the viruses (Figure 1 and Appendix A). The presence of ISVs was then verified with RT-PCR (Appendix A).

### 3.2. Genomic Structures Analysis of Three Novel Viruses of R. pedestris

Among the seven novel RNA viruses identified, three that had complete ORF of RNA-dependent RNA polymerase (RdRp) and nearly full-length genome sequences were selected for further analysis, including Riptortus pedestris Alphadrosrha-like virus 1 (RpALv1), Riptortus pedestris Alphadrosrha-like virus 2 (RpALv2) and Riptortus pedestris Almendra-like virus (RiALv). Based on the results of BLASTx, RdRp aa of RpALv1 and RpALv2 shared 30.28% and 30.38% aa identities with the reference sequences Hymenopteran rhabdo-related virus OKIAV24 (accession:MW039260.1) and Wuhan House Fly Virus 2 (accession:NC_031283), while RiALv shared 33.58% aa identity with soybean thrips rhabdo-like virus 2 (accession:MT224148). According to the closely related homology viruses, all these three newly identified viruses are potentially belonging to the family Rhabdoviridae (Table 1). To further investigate genome structures of the novel viruses, open reading frame (ORF) were predicted with NCBI ORF Finder. The results showed that RpALv1 and RpALv2 contained 6 ORFs (Figure 1A,B), whereas RiALv had 5 ORFs (Figure 1C). According to the InterProScan-based prediction of conserved domains, the ORF1 of RpALv1 and RiALv contained the conserved domains of Rhabdovirus nucleocapsid proteins (Ncap) (Figure 1A,C). Importantly, the largest ORF of all the three novel viruses, includes Mononegavirales RNA dependent RNA polymerase (Mononeg_RNA_pol), Mononegavirales mRNA-capping region V (Mononeg_mRNAcap) and Virus-capping methyltransferase (Mononega_L_MeTrfase) (Figure 1).

### 3.3. Phylogenetic Analysis of Three Novel Viruses of R. pedestris

To determine the taxonomic status of the three novel viruses, phylogenetic analysis was performed based on the aa sequence of RdRp using ML methods. The two novel viruses RpALv1 and RpALv2 were clustered together in a distinct clade and were separated from those viruses in the genera Alphadrosrha and Betanemeha, which were supported by high bootstrap values (Figure 2A). Based on the new genus demarcation criteria of Rhabdoviridae by ICTV [38], it is speculated that RpALv1 and RpALv2 may belong to a new genus in this family. Similarly, ML tree suggested that RiALv is also separated from those viruses in known genera, as supported by a relatively high bootstrap value (Figure 2B). Thus, phylogenetic analysis clearly showed that the three newly identified viruses (RpALv1, RpALv2 and RiALv), possibly belonging to new genera, are the new members in the family Rhabdoviridae.

### 3.4. Virus-Derived Small Interfering RNA (vsiRNA) Analysis of R. pedestris Responsive to Virus Infection

In insects, small-interfering RNA (siRNA)-based RNA interference (RNAi) is an important antiviral pathway against viral invasion, and the activation of the RNAi antiviral pathway is usually evaluated by the accumulation of virus-derived siRNAs (vsiRNAs) [39]. To explore the siRNA-based antiviral immunity in *R. pedestris*, sRNA of *R. pedestris* were sequenced and vsiRNAs were further analyzed. As shown in Figure 3, the vsiRNAs derived from RiALv were mostly 22 nt in length and equally derived from the sense and antisense strands of the viral genome. Moreover, the 22 nt long vsiRNAs of RiALv had a strong A/U bias in the 5′-terminal nucleotides (Figure 3). The typical characteristics of vsiRNAs strongly suggested that RiALv might successfully replicate in *R. pedestris*, and the RNAi antiviral pathway of *R. pedestris* was actively involved in the response to RiALv infections. By contrast, it was unexpected that no typical distribution and features of vsiRNA were observed for RpALv1 and RpALv2 (Appendix A), although the viral coverage of the two viruses were relatively high in *R. pedestris* (Figure 1A,B).

## 4. Discussion

With the development of high-throughput sequencing technology, the discovery of ISVs has entered a new era [16], and numerous ISVs have been identified from many insects of the order Hemiptera. For example, the whitefly *Bemisia tabaci*, a well-known insect vector, has been found to contain more than 20 ISVs, most of which are unclassified [28]. However, limited ISVs have been reported in *R. pedestris* to date. In this study, ten ISVs were identified in *R. pedestris*, seven of which were novel viruses (Table 1). Notably, Guiyang srgiope bruennichi iflavirus 1, one of the identified ISVs in *R. pedestris*, was previously reported in the spider host *Argiope bruennichi*. Considering the potential predation relationship between spider and bean bug [40,41], as well as the limitations of metatranscriptomics approaches to determining the host of ISVs [42], it is speculated that *R. pedestris* may be the authentic host of this virus.

Here, this work focused on the three novel viruses RpALv1, RpALv2 and RiALv with complete/near complete genomes, which belonged to the family Rhabdoviridae. Rhabdovirus has a negative-sense single-stranded RNA genome (-ssRNA) that contains five canonical structural genes [43,44]. Among them, the first structural protein (nucleocapsid protein) contains the Ncap domain, which was also predicted in RpALv1 and RiALv (Figure 1A,C) but not in RpALv2 (Figure 1B). The reason might be because the overlap of other accessory genes or gene mutations interferes with the prediction results. Similar results were also observed in other viruses, such as Coot Bay virus, where some residues of the nucleocapsid protein are mutated [45], indicating that the structures of the nucleocapsid proteins are highly diverse and non-conserved. Furthermore, these three novel viruses are clustered separately from other rhabdoviruses, possibly new genera in this family (Figure 2).

In insects, siRNA-based RNA silencing is an important antiviral pathway. In this study, most vsiRNAs derived from RiALv were 22 nt long and equally derived from the sense and antisense strands of the viral genome RNA (Figure 3). These results were similar to previous reports [27,46] indicating that the 22 nt vsiRNA peak was commonly observed for the insect hosts in the order Hemipterans, and vsiRNAs of RiALv might be processed in a same way in bean bugs. Nevertheless, no typical features of vsiRNA were observed in RpALv1 and RpALv2, indicating that RpALv1 and RpALv2 might not elicit an RNAi antiviral response. Consequently, it is speculated that the siRNA-based antiviral immunity of *R. pedestris* might be more complex than previously thought which needs further investigation. It will be interesting to further explore the details of siRNA-based antiviral immunity in bean bugs in response to various types of virus infection.

## 5. Conclusions

In summary, three novel viruses, including RpALv1, RpALv2 and RiALv, were identified and chracterized in *R. pedestris* originating from different regions using metatranscriptome sequencing technology. Phylogenetic tree analysis based on the aa sequences of RdRp revealed that three novel viruses were clustered into new distinct clades in the family Rhabdoviridae. Collectively, the RNA virome revealed in this study contributes to the better understanding for the virus diversity in *R. pedestris* and provides ISV resources for the potential development of soybean bug-specific biological control methods.

## Figures and Tables

**Figure 1 viruses-14-02500-f001:**
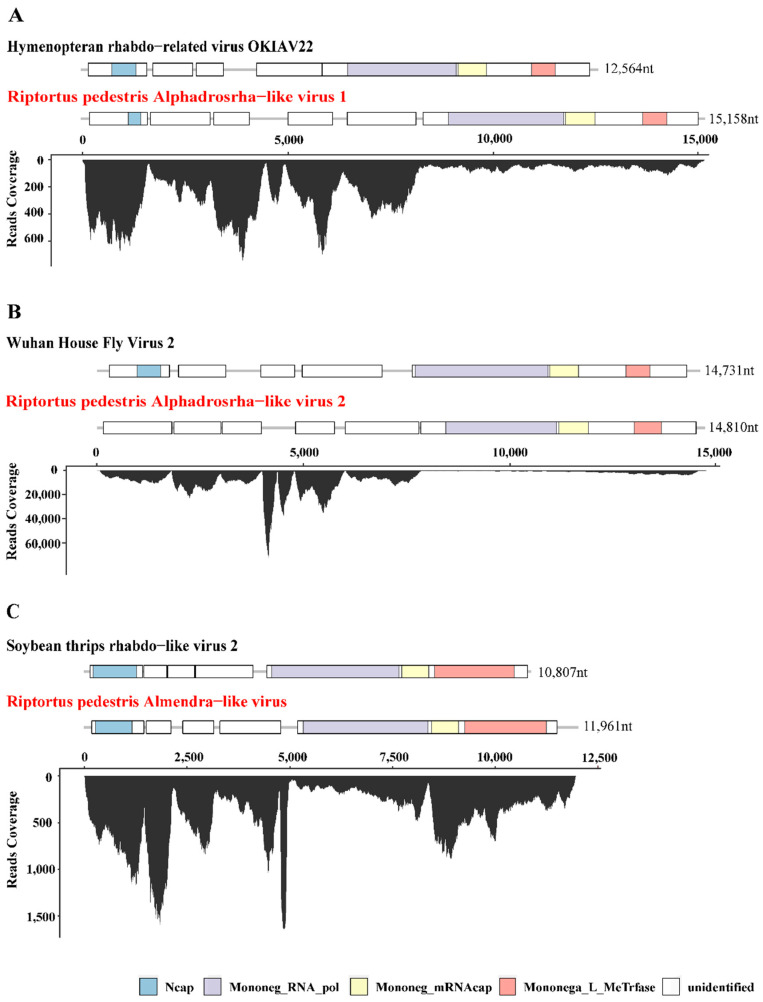
Genome structure and transcriptome raw read coverage of three novel insect-specific viruses identified in *R. pedestris.* The viruses were divided into 3 groups as shown in panels (**A**–**C**). Each panel contains two viruses, the reference genome with high homology on the upper panel (with black font) and the novel virus identified in *R. pedestris* on the lower panel (with red font). Each box represents an open reading frame (ORF) of the viruses. Conserved functional domains are color-coded and the corresponding names were shown at the bottom of the figure. Abbreviations of the conserved domain names: Ncap, Rhabdovirus nucleocapsid proteins; Mononeg_RNA_pol, Mononegavirales RNA dependent RNA polymerase; Mononeg_mRNAcap, Mononegavirales mRNA-capping region V; Mononega_L_MeTrfase, Virus-capping methyltransferase.

**Figure 2 viruses-14-02500-f002:**
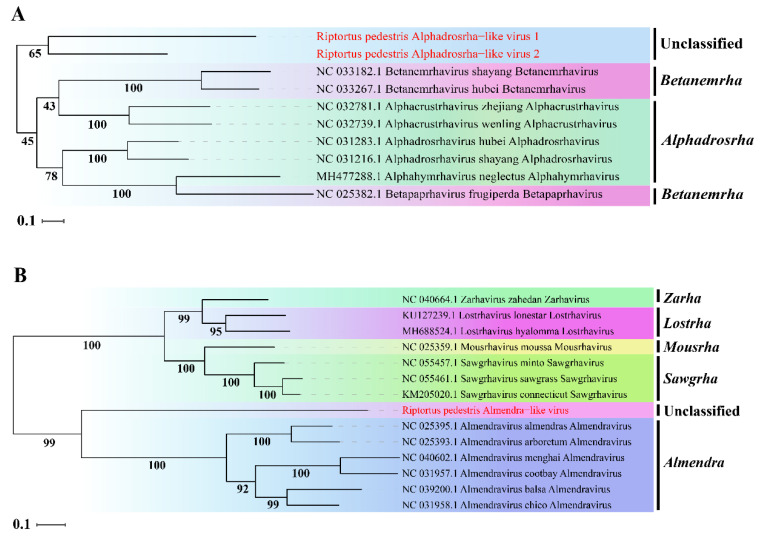
ML-phylogenetic tree of three insect-specific viruses identified in *R. pedestris*. Phylogenetic tree based on the RdRp domain of Riptortus pedestris alphadrosrha-like virus 1 (RpALv1), Riptortus pedestris alphadrosrha-like virus 2 (RpALv2) (**A**) and Riptortus pedestris almendra-like virus (RiALv) (**B**) were constructed using maximum likelihood method. ISVs identified in this study are shown in red font. Nodes with bootstrap values >50% are placed over each node of the tree. Scale bars on the left represent percentage divergence.

**Figure 3 viruses-14-02500-f003:**
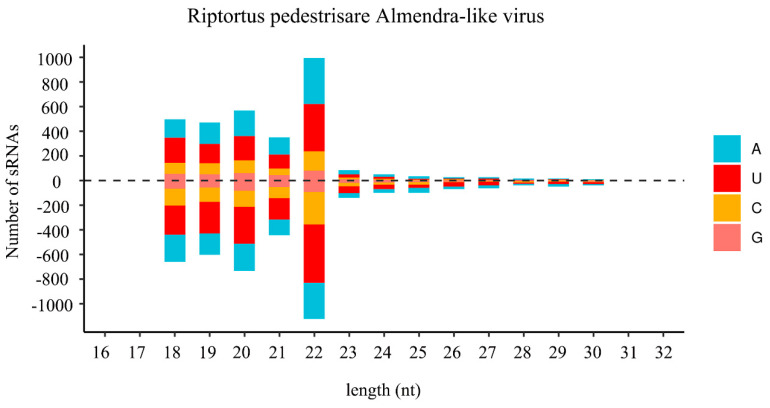
Profile of virus-derived small interfering RNAs (vsiRNAs) of Riptortus pedestris almendra-like virus (RiALv). Positive and negative values indicate that the vsiRNA was derived from the positive- and negative-sense strands. Color coding shows different nucleotides, and the corresponding names are list on the right.

**Table 1 viruses-14-02500-t001:** Insect-specific viruses identified in *R. pedestris*.

Temporary Virus Name	Collection Area	Sequence Length (nt)	Coverage	E-Value	Homologous Virus (Genome Size, nt)	Protein Identities	Virus Family	Virus Genus
Riptortus pedestris virus-1 Isolate AH-SZ	AH-SZ	10,429	17×	0.0	Riptortus pedestris virus-1 (10,527)	96.29%	*Unassigned*	*Unassigned*
Riptortus pedestris virus-1 Isolate AH-HF	AH-HF	10,492	13×	0.0		96.30%		
Riptortus pedestris virus-1 Isolate AH-FY	AH-FY	10,161	16×	0.0		96.30%		
Guiyang srgiope bruennichi iflavirus 1 Isolate SD-HZ	SD-HZ	7711	74×	0.0	Guiyang srgiope bruennichi iflavirus 1 (10,414)	94.64%	*Iflaviridae*	*Iflavirus*
Guiyang srgiope bruennichi iflavirus 1 Isolate SD-JN	SD-JN	8260	66×	0.0		94.75%		
Riptortus pedestris virus-2 Isolate SD-JN	SD-JN	10,021	8151×	0.0	Riptortus pedestris virus-2(9915)	98.90%	*Unassigned*	*Unassigned*
Riptortus pedestris Reoviridae-like virus (RpRLv)	AH-SZ	2249	13×	2 × 10^−48^	Nephila clavipes virus 6 (9506)	26.12%	*Reoviridae*	*Unassigned*
***** Riptortus pedestris Alphadrosrha-like virus 1 (RpALv1)	SD-JN	15,158	216×	3 × 10^−146^	Hymenopteran rhabdo-related virus OKIAV24(12,564)	30.28%	*Rhabdoviridae*	*Unassigned*
***** Riptortus pedestris Alphadrosrha-like virus 2 (RpALv2)	SD-JN	14,810	7295×	0.0	Wuhan House Fly Virus 2 (14,731)	30.38%	*Rhabdoviridae*	*Unassigned*
Riptortus pedestris Permutotetraviridae-like virus (RpPLv)	SD-JN	5399	60×	0.0	Sanya permutotetra-like virus 3(5461)	40.64%	*Permutotetraviridae*	*Unassigned*
***** Riptortus pedestris Almendra-like virus (RiALv)	AH-XC	11,961	474×	0.0	Soybean thrips rhabdo-like virus 2 (10,807)	33.58%	*Rhabdoviridae*	*Unassigned*
Riptortus pedestris Flaviviridae-like virus 1 (RpFLv1)	SD-JN	3667	14×	8 × 10^−177^	Changjiang Jingmen-like virus (3127)	44%	*Flaviviridae*	*Unassigned*
Riptortus pedestris Flaviviridae-like virus 2 (RpFLv2)	SD-JN	2585	11×	2 × 10^−91^	Wuhan flea virus (2863)	32.79%	*Flaviviridae*	*Unassigned*

***** The novel insect-specific viruses identified in *R. pedestris*.

## Data Availability

The raw reads of RNA-seq generated in this study were deposited in NCBI SRA with accession numbers SRR21672252 (AH-A1), SRR21672251 (AH-A2), SRR21672250 (AH-A3), SRR21672249 (AH-A4), SRR21672248 (SD-B1), SRR21672247 (SD-B2) for transcriptome, and SRR21673402 (AH-A1), SRR21673401 (AH-A2), SRR21673400 (AH-A3), SRR21673399 (AH-A4), SRR21673398 (SD-B1), SRR21673397 (SD-B2) for sRNA, respectively. The full genome sequences of viruses in this study have been deposited into GenBank with the accession number OP177939 (Riptortus pedestris virus-1 Isolate AH-SZ); OP177940 (Riptortus pedestris virus-1 Isolate AH-HF); OP177941 (Riptortus pedestris virus-1 Isolate AH-FY); OP328360 (Guiyang srgiope bruennichi iflavirus 1 Isolate SD-HZ); OP328361 (Guiyang srgiope bruennichi iflavirus 1 Isolate SD-JN); OP328362 (Riptortus pedestris virus-2 Isolate SD-JN); OP589936 (Riptortus pedestris Reoviridae-like virus); OP589942 (Riptortus pedestris Alphadrosrha-like virus (1); OP589940 (Riptortus pedestris Alphadrosrha-like virus (2); OP589937 (Riptortus pedestris Permutotetraviridae-like virus); OP589941 (Riptortus pedestris Almendra-like virus); OP589938 (Riptortus pedestris Flaviviridae-like virus 1); OP589939 (Riptortus pedestris Flaviviridae-like virus (2).

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
