# Peer review of "The Characterization of Three Novel Insect-Specific Viruses Discovered in the Bean Bug, Riptortus pedestris"

_viruses, 2022, doi:10.3390/v14112500_

Round 1

Reviewer 1 Report

In this manuscript, the authors investigated the insect-specific virus from R. pedestris in China and three new insect RNA viruses were identified and chracterized using metatranscriptome sequencing technology. This might be interesting topic for publication in this special issue. The list of the corrected points described below, but especially the introduction and discussion should be reconsidered.

[Abstract]

Line 23-24: It is unclear the effect of ISVs identified in this study, and its potential as biological control agents needs further evaluation.

[Introduction]

Line 40-41 Please list the citation.

Line 41-44 On the evidence so far, the stinkbug is just one of the factors that cause the soybean staygreen syndrome. The authors should cite and state the information about the soybean staygreen syndrome, and discuss considering both this insect and plant viruses.

(1) Characterization of Salivary Secreted Proteins That Induce Cell Death From Riptortus pedestris (Fabricius) and Their Roles in Insect-Plant Interactions. Frontiers in Plant Science. https://doi.org/10.3389/fpls.2022.912603

(2) Field Cage Assessment of Feeding Damage by Riptortus pedestris on Soybeans in China. Insects. https://doi.org/10.3390/insects12030255

(3) A new distinct geminivirus causes soybean stay-green disease. Molecular Plant. https://doi.org/10.1016/j.molp.2022.03.011

Identification and characterization of a new geminivirus from soybean plants and determination of V2 as a pathogenicity factor and silencing suppressor. BMC Plant Biology. https://doi.org/10.1186/s12870-022-03745-z

(4) Characterization of Salivary Secreted Proteins That Induce Cell Death From Riptortus pedestris (Fabricius) and Their Roles in Insect-Plant Interactions. Frontiers in Plant Science. DOI: 10.3389/FPLS.2022.912603

Line57-58 The authors should provide more information about Riptortus pedestris virus 1 (RiPV1) and Riptortus pedestris virus 2 (RiPV2).

[Materials and Methods]

Line 77 The developmental stage and physiological condition of R. pedestris used in this study should be described in more detail, e.g., insect body parts, day-old, copulation or not.

[Discussion]

Line 226- 228 Please list the citation.

Does ISV identified in this study comes from field environment or indoor rearing environment?

Is there any connection between ISVs identified in this study and soybean staygreen syndrome?

What is the Innovative Points of this study compared to the authors' reference [21-22]?

Please cite and state the information.

Author Response

Response to Reviewer 1’s comments:

Abstract

1- Line 23-24: It is unclear the effect of ISVs identified in this study, and its potential as biological control agents needs further evaluation.

Reply: Thanks for your valuable comments. The description has been revised as follow: “In conclusion, this study contributes to a better understanding of the microenvironment in R. pedestris and provides viral information for the development of potential soybean insect-specific biocontrol agents.” (Please see the revised manuscript on Page 1 Line 23 to Line 25)

Introduction

2-Line 40-41: Please list the citation

Reply: Added as suggested. (Please see the revised manuscript on Page 1 Line 42)

Reference:

  1. Feeding of Riptortus pedestris on soybean plants, the primary cause of soybean staygreen syndrome in the Huang-Huai-Hai river basin. Crop J. 2019, 7, 360-367.

3- Line 41-44: On the evidence so far, the stinkbug is just one of the factors that cause the soybean staygreen syndrome. The authors should cite and state the information about the soybean staygreen syndrome, and discuss considering both this insect and plant viruses.

Reply: Thanks for your suggestion. As suggested, we described this paragraph as follow: “Currently, ‘Zhengqing’ has become the most epidemic and prominent issue in the Huang-Huai-Hai soybean producing area of China [1], and the direct cause of ‘Zhengqing’ is still highly debated. Although ‘Zhengqing’ can be induced by the pod removal or seed injury [8], recent studies demonstrated that a new geminivirus is also associated with soybean staygreen syndrome [9-11]. Additionally, field cage experiments demonstrated that R. pedestris infestation of soybean can clearly induce the increased numbers of abnormal pods, aborted seeds, staygreen leaves, and decreased yields [1,12].” (Please see the revised manuscript on Page 1 Line 42 to Line 47)

Reference:

  1. Cheng, R.; Mei, R.; Yan, R.; Chen, H.; Miao, D.; Cai, L.; Fan, J.; Li, G.; Xu, R.; Lu, W.; et al. A new distinct geminivirus causes soybean stay-green disease. Mol. Plant 2022, 15, 927-930.
  2. Wang, X.; Wang, M.; Wang, L.; Feng, H.; He, X.; Chang, S.; Wang, D.; Wang, L.; Yang, J.; An, G.; et al. Whole-plant microbiome profiling reveals a novel geminivirus associated with soybean stay-green disease. Plant Biotechnol. J. 2022. doi.org/10.1111/pbi.13896
  3. Li, Q.; Zhang, Y.; Lu, W.; Han, X.; Yang, L.; Shi, Y.; Li, H.; Chen, L.; Liu, Y.; Yang, X.; et al. Identification and characterization of a new geminivirus from soybean plants and determination of V2 as a pathogenicity factor and silencing suppressor. BMC Plant Biol. 2022, 22, 362.
  4. Li, W.; Gao, Y.; Hu, Y.; Chen, J.; Zhang, J.; Shi, S. Field cage assessment of feeding damage by Riptortus pedestris on soybeans in China. Insects 2021, 12, 255.

4-Line57-58: The authors should provide more information about Riptortus pedestris virus-1 (RiPV-1) and Riptortus pedestris virus-2 (RiPV-2).

Reply: As suggested, detailed description about RiPV-1 and RiPV-2 has been added as follow: “However, few ISVs have been reported in R. pedestris, and presently, the only ISVs reported in R. pedestris are Riptortus pedestris virus 1 (RiPV-1) and Riptortus pedestris virus 2 (RiPV-2), which were discovered from R. pedestris that infected with the entomopathogenic fungus Beaveria bassiana [25,26]. RiPV-1 and RiPV-2 are positive-sense single-stranded RNA viruses belonging to the order Picornavirales. The genomes of both viruses are 9-11 kb in length and contain a large Open Reading Frame (ORF) that encodes multiple conserved motifs, including helicase, RNA-dependent RNA polymerase(RdRP), etc [25,26].” (Please see the revised manuscript on Page 2 Line 67 to Line 72)

Materials and Methods

  • Line 77: The developmental stage and physiological condition of pedestris used in this study should be described in more detail, e.g., insect body parts, day-old, copulation or not.

Reply: Thanks for the valuable comments. As suggested, detailed description has been added in the “RNA Extraction” section. It is as follow: “Total RNA was extracted from insects by using Trizol reagent (Invitrogen, USA) following the manufacturer's instructions. Briefly, a pair of adult R. pedestris was randomly collected from the soybeans of each area and then the whole bodies of the alive insects were moved into RNase-free microfuge tubes (2.0 mL). Subsequently, the extracted RNA samples were equally divided into two tubes for transcriptome and small RNA (sRNA) sequencing, respectively.” (Please see the revised manuscript on Page 2 Line 92 to Line 94)

Discussion

  • Line 226- 228: Please list the citation.

Reply: Added as suggested. (Please see the revised manuscript on Page 9 Line 243 to 246)

Reference:

  1. Szymkowiak, P.; Tryjanowski, P.; Winiecki, A.; Grobelny, S.; Konwerski, S. Habitat differences in the food composition of the wasplike spider Argiope bruennichi (Scop.) (Aranei: Araneidae) in Poland. Belg. J. Zool. 2005, 135, 33-37.
  2. Ludy, C. Prey selection of orb-web spiders (Araneidae) on field margins. Agric. Ecosyst. Environ. 2007, 119, 368-372.
  3. Cobbin, J. C.; Charon, J.; Harvey, E.; Holmes, E. C.; Mahar, J. E. Current challenges to virus discovery by meta-transcriptomics. Curr. Opin. Virol. 2021, 51, 48-55.
  • Does ISV identify in this study comes from field environment or indoor rearing environment?

Reply: Thanks for your valuable comments and we are sorry for the confusion. ISVs identified in this study were all derived from field insects. Detailed description has been added in the “Field Insect Samples” section. It is as follow: “The insect was sent to our laboratory alive, some adult R. pedestris were transferred to trizol reagent (Invitrogen, USA) for RNA extraction, and remaining insect samples were reared separately in nylon mesh cages with cotton pads soaked in water and potted soybean plants (cv. Qihuang 34) in Ningbo University, Zhejiang.” (Please see the revised manuscript on Page 2 Line 85 to Line 86)

  • Is there any connection between ISVs identified in this study and soybean staygreen syndrome?

Reply: Thanks. In this study, we did not observe significant correlation between identified ISVs and soybean staygreen syndrome in the field. To the best of our knowledge, currently, there are no reports indicated the relationship between ISVs and soybean staygreen syndrome. The potential correlation between ISVs and soybean staygreen syndrome will be interesting topic to be elucidated in the future studies.

  • What is the Innovative Points of this study compared to the authors' reference [21-22]?

Reply: Thanks for your valuable comments. Previously, the R. pedestris used to identify the two viruses (RiPV-1 [25] and RiPV-2 [26]) were infected with the entomopathogenic fungus Beaveria bassiana, and they were focused on gene regulation during fungal infection and the interaction between virus, fungus and the host.

In this study, insect samples were collected from various soybean planting areas in China, and multiple ISVs were discovered. Moreover, small interfering RNA (siRNA) experiments were performed to confirm the host's sirna-based antiviral capabilities, indicating the replication of these ISVs in the insects. Our study mainly aimed to provid ISV diversities in the field R. pedestris which might contribute to the future biocontrol of this pest.

Reviewer 2 Report

The authors collected Riptortus pedestris from six soybean fields located in China. They studied the virome of these insects using the metatranscriptomic technology. Furthermore, they conducted a vsiRNA analysis. They identified in total 10 viruses of which seven are new insect RNA viruses. However, only three out of the seven viruses had almost a complete genome sequence identified in their RNA-Seq data. The phylogenetic analysis of these three viruses’ RNA-dependent RNA polymerase (RdRp) gene revealed that the three viruses are clustered into new distinct clades. Overall, the manuscript is well organised and follow scientifically sound experimental methods.

Comments:

In methods section, the performance of other techniques such as RT-PCR and Sanger sequencing especially for the newly identified viral genomes will validate the findings of this study.

In results section (3.4), the last paragraph belongs to the discussion section.

In discussion section page 8, lines (225-228), I am not sure of the meaning, please rewrite the sentence and add reference(s).

Minor corrections:

Riptortus pedestris should be italic>>check in ALL the manuscript.

Page 8, Line 221>> Bemisia tabaci should be italic.

Page 9, Line 264>> the word “collect” should be “collected”.

Page 9, Line 261>> a space is missing before the word “Figure S2”.

Pages 9, 10, 11 (reference section)>> Please revise ALL the scientific names of insects, they should ALL be written in italic.

Author Response

Response to Reviewer 2’s comments:

1-In methods section, the performance of other techniques such as RT-PCR and Sanger sequencing especially for the newly identified viral genomes will validate the findings of this study.

Reply: Thanks for the constructive suggestions. As suggested, we have performed the RT-PCR with ISV specific primers to identify the presence of these RNA viruses in R. pedestris. The RT-PCR results and the ISV specific primers were provided as New Supplementary Figure S2 and Supplementary Table S1. The description of RT-PCR methods and results were added in the “Materials and Methods 2.4” section (Please see the revised manuscript on Page 3 Line 108 to Line 111), and “Results 3.1” section. (Please see the revised manuscript on Page 4 Line 162)

2-In results section (3.4), the last paragraph belongs to the discussion section.

Reply: As suggested, we have removed this paragraph to the “Discussion” section in the revised manuscript. (Please see the revised manuscript on Page 10 Line 266 to Line 268)

3-In discussion section page 8, lines (225-228), I am not sure of the meaning, please rewrite the sentence and add reference(s).

Reply: As suggested, we rewrote this sentence in “Discussion” section. It is as follow: “Notably, Guiyang srgiope bruennichi iflavirus 1, one of the identified ISVs in R. pedestris, was previously reported in the spider host Argiope bruennichi. Considering the potential predation relationship between spider and beanbug [40,41], as well as the limitations of metatranscriptomics approaches to determine the host of ISVs [42], it is speculated that R. pedestris may be the authentic host of this virus.” (Please see the revised manuscript on Page 9 Line 243 to Line 246)

Reference:

  1. Szymkowiak, P.; Tryjanowski, P.; Winiecki, A.; Grobelny, S.; Konwerski, S. Habitat differences in the food composition of the wasplike spider Argiope bruennichi (Scop.) (Aranei: Araneidae) in Poland. Belg. J. Zool. 2005, 135, 33-37.
  2. Ludy, C. Prey selection of orb-web spiders (Araneidae) on field margins. Agric. Ecosyst. Environ. 2007, 119, 368-372.
  3. Cobbin, J. C.; Charon, J.; Harvey, E.; Holmes, E. C.; Mahar, J. E. Current challenges to virus discovery by meta-transcriptomics. Curr. Opin. Virol. 2021, 51, 48-55.

4-Riptortus pedestris should be italic>>check in ALL the manuscript.

Reply: Corrected as suggested.

5-Page 8, Line 221>> Bemisia tabaci should be italic.

Reply: Corrected as suggested.

6-Page 9, Line 264>> the word “collect” should be “collected”.

Reply: Corrected as suggested.

7-Page 9, Line 261>> a space is missing before the word “Figure S2”.

Reply: Corrected as suggested.

8-Pages 9, 10, 11 (reference section)>> Please revise ALL the scientific names of insects, they should ALL be written in italic.

Reply: Corrected as suggested.

Round 2

Reviewer 1 Report

The manuscript has improved a lot from the previous version. The introduction provide sufficient background and include all relevant references. I think the paper can be accepted in present form. 

Author Response

We are very grateful for your support and expert comments on this work.